# Medical management of muscle weakness in Duchenne muscular dystrophy

**Sarah R. Rivera** [1] *, **Sumit K. Jhamb** [2], **Hoda Z. Abdel-Hamid** [3‡], **Gyula Acsadi** [4‡], **John Brandsema** [5‡], **Emma Ciafaloni** [6‡], **Basil T. Darras** [7‡], **Susan T. Iannaccone** [8‡], **Chamindra G. Konersman** [9‡], **Nancy L. Kuntz** [10‡], **Craig M. McDonald** [11‡], **Julie A. Parsons** [12‡], **Carolina Tesi Rocha** [13‡], **Craig M. Zaidman** [14‡], **Russell J. Butterfield** [15☯], **Anne M. Connolly** [16☯], **Katherine D. Mathews** [17☯]

1 Department of Clinical Services, Optum Lifesciences Wolfeboro, Wolfeboro, New Hampshire, United States of America, 2 Department of Clinical Services, Optum Global Solutions, Noida, Uttar Pradesh, India, 3 Division of Child Neurology, Children's Hospital of Pittsburgh of UPMC, Pittsburgh, Pennsylvania, United States of America, 4 Department of Neurology, Connecticut Children's Medical Center, Farmington, Connecticut, United States of America, 5 Division of Neurology, Children's Hospital of Philadelphia, Perelman School of Medicine at the University of Pennsylvania, Philadelphia, Pennsylvania, United States of America, 6 Department of Pediatric Neuromuscular Medicine, University of Rochester Medical Center, Rochester, New York, United States of America, 7 Department of Neurology, Boston Children's Hospital and Harvard Medical School, Boston, Massachusetts, United States of America, 8 Department of Pediatrics, UT Southwestern, Dallas, Texas, United States of America, 9 Department of Neurosciences, University of California, San Diego, San Diego, California, United States of America, 10 Department of Pediatrics, Ann & Robert H Lurie Children's Hospital, Chicago, Illinois, United States of America, 11 Department of Physical Medicine & Rehabilitation, UC Davis Health, Sacramento, California, United States of America, 12 Department of Pediatrics and Neurology, University of Colorado School of Medicine, Aurora, Colorado, United States of America, 13 Department of Neurology, Stanford University, Palo Alto, California, United States of America, 14 Department of Neurology, Divisions of Child Neurology and Neuromuscular, Washington, University in St. Louis School of Medicine, St. Louis, Missouri, United States of America, 15 Department of Neurology and Pediatrics, University of Utah, Salt Lake City, Utah, United States of America, 16 Department of Neurology, Nationwide Children's Hospital, Columbus, Ohio, United States of America, 17 Departments of Pediatrics, University of Iowa Carver College of Medicine, Iowa City, Iowa, United States of America

☯ These authors contributed equally to this work.
‡ These authors also contributed equally to this work
* riverapharmd@gmail.com

**Data Availability Statement:** All relevant data are within the manuscript and Supporting Information files.

## Abstract

### Introduction

Duchenne muscular dystrophy (DMD) is a childhood onset muscular dystrophy leading to shortened life expectancy. There are gaps in published DMD care guidelines regarding recently approved DMD medications and alternative steroid dosing regimens.

### Methods

A list of statements about use of currently available therapies for DMD in the United States was developed based on a systematic literature review and expert panel feedback. Panelists' responses were collected using a modified Delphi approach.

### Results

Among corticosteroid regimens, either deflazacort or prednisone weekend dosing was preferred when payer requirements do not dictate choice. Most patients with exon 51 skip-amenable mutations should be offered eteplirsen, before or with a corticosteroid.

**Funding:** Sarepta funded this study. The funder Optum provided support in the form of salaries for authors SRR and SKJ but did not have any additional role in the study design, data collection and analysis, decision to publish, or preparation of the manuscript. The specific roles of these authors are articulated in the 'author contributions' section.

**Competing interests:** SRR and SKJ are currently employed by Optum. Sarepta funded this study. This does not alter our adherence to PLOS ONE policies on sharing data and materials.

## Discussion

The options available for medical management of the motor symptoms of DMD are expanding rapidly. The choice of medical therapies should balance expected benefit with side effects.

## Introduction

Duchenne muscular dystrophy (DMD) is an X-linked recessive muscle disorder resulting in progressive weakness, loss of ambulation, and premature death due to respiratory and cardiac failure [1–4]. Historically, patients with DMD did not survive past late teens or early 20s [1,5,6]. With current management, life expectancy has been extended by 5–15 years [5,7]. The management of DMD is complex and requires a multidisciplinary approach. The DMD Care Considerations, funded by the Centers for Disease Control and Prevention (CDC), provided expert opinion, and evidence based guidelines for the diagnosis, management, and care continuum of DMD [8–10]. However, since these guidelines were published, new medications have received FDA approval for treatment of DMD. Here, we present expert opinion on the use of newly approved medications in DMD to assist in clinical decision-making not addressed in the published Care Considerations.

## Methods

A steering committee for this project was comprised of three experts specialized in the diagnosis and treatment of DMD (K. D. Mathews, chair, A. M. Connolly, and R. J. Butterfield; combined 62 years of experience in DMD care at the faculty level). This steering committee participated in all aspects and conduct of this study. Co-authors (HZA, GA, JB, EC, BTD, STI, CGK, NLK, CMM, JAP, CTR, and CMZ) together with the steering committee members comprise the expert panel who contributed to the consensus.

Gaps in current treatment guidance for the medical management of muscle weakness in DMD were identified by literature review, with additional input by the steering committee. Based on identified gaps, a list of statements was developed for panelist response using a modified Delphi method as summarized in Fig 1. The expert panel was surveyed using an online portal "Survey Monkey". Most statements were addressed with a 5-point scale from strongly agree to strongly disagree. The statements regarding corticosteroids focused on three regimens commonly used in the United States and for which published data exists: daily prednisone (DP), weekend prednisone (WP) and daily deflazacort (DD). Questions focused on expert opinion regarding best regimen for treatment and what issues should be considered in choosing a corticosteroid regimen. The statements regarding eteplirsen focused on which patients should be offered treatment with eteplirsen, how treatment response should be monitored, and safety of eteplirsen.

The first two rounds of interviews/survey were electronic only, to ensure anonymity and unbiased reponses. During the first round of survey, the panel rated each statement based on their clinical experience, knowledge, and current practice patterns for the management of DMD. The panel members also provided comments regarding the inclusion/exclusion of the statements as well as the clarity of the statements to understand and make informed decisions. Following this, the statements were updated based on feedback from the panel to improve clarity. The second-round of surveys was conducted by using revised or new statements agreed upon by the steering committee. Next, a third survey was conducted via a series of teleconferences to identify gaps not yet addressed and areas where greater clarity was required. Based on

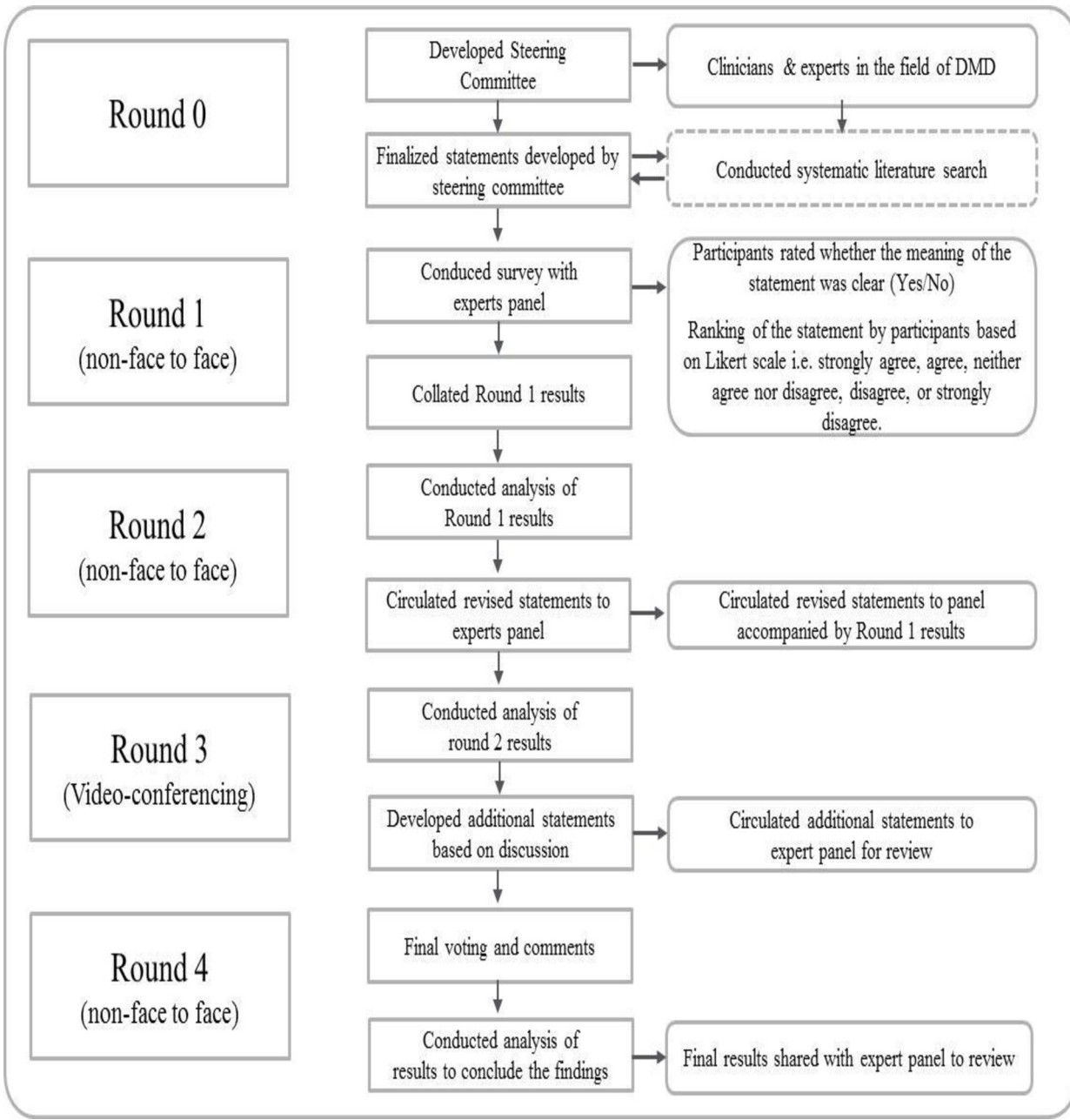

**Fig 1. Overview of consensus statement development using the modified Delphi method.**

this, a final fourth survey was conducted to ensure that the panel members' reasoning and consensus was adequately captured. The data collection through surveys and teleconference was carried out between 13 November 2019 and 4 December, 2019. For analysis, the statements were grouped "agree" with "strongly agree", and "disagree" with "strongly disagree".

## Results

All 15 panel members voted on each round of the survey and participated in teleconferences. The responses from all rounds of survey were collated for analysis and the final responses are

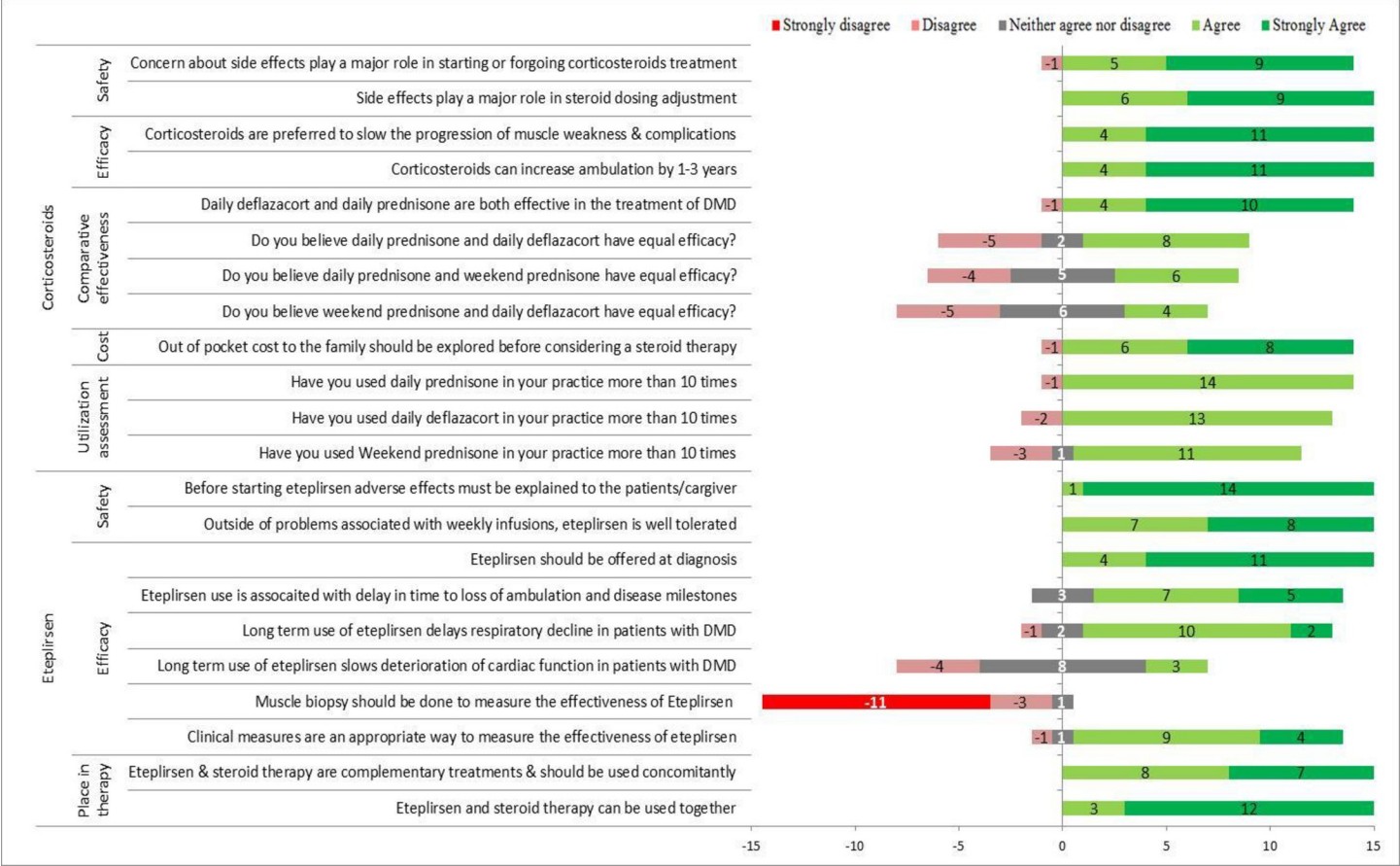

**Fig 2. Consensus level on the key statements rated by expert panel.**

summarized in Fig 2. The specific statements and voting results for all the rounds are presented as S1 Table (survey round 1) and S2 and S3 Tables (survey round 2&4).

## Corticosteroids

Statements and responses regarding use of corticosteroids are summarized in Fig 2, along with detailed responses in S2 Table. There was 100% consensus that corticosteroids are effective in DMD. There was no clear consensus about relative effectiveness or clinical preference for the three corticosteroid regimens explored: DP, WP and DD. Most panel members have used all three dosing regimens in practice (Fig 2 and Table 1, Statement 1) and felt that all three approaches were acceptable regimens. The clinician perception of comparative effectiveness is summarized in Fig 2. When asked specifically about the strength of evidence supporting each of the regimens, panel members responded that there was evidence to support all three regimens. However, 10 respondents strongly agreed that there was strong evidence for the daily dosing (prednisone and deflazacort respectively) compared to 3 strong agreement responses for WP (S2 Table, Statement 7).

If payer requirements were not a consideration, panalists were divided on the most appropriate dosing regimen between DD (eight panelists) and WP (six panelists) (Table 1, Statement 2). Daily prednisone was favored as a first line medication by only one panel member.

**Table 1. Experts opinion regarding utilization and preference over different corticosteroid regimens (survey round 2&4).**

| Statement 1: Utilization assessment | > 10 times | <10 times | Not at all |
|---|---|---|---|
| a. Have you used daily prednisone in your practice? | 14 | 1 | 0 |
| b. Have you used weekend prednisone in your practice? | 11 | 3 | 1 |
| c. Have you used daily deflazacort in your practice? | 13 | 2 | 0 |
| **Statement 2: Treatment preferences** | **DP** | **WP** | **DD** |
| If payer requirements were not an issue what would be your first choice for corticosteroid treatment? | 1 | 6 | 8 |
| **Statement 3: For patients where steroid use is clinically appropriate, please rank the following steroid treatments based on your preference.** | **DP** | **WP** | **DD** |
| a. Rank 1 (no. of responses) | 3 | 7 | 7 |
| b. Rank 2 (no. of responses) | 9 | 2 | 6 |
| c. Rank 3 (no. of responses) | 3 | 6 | 2 |

Side effects differ among corticosteroid dosing regimens.There was consensus that corticosteroid side effects influence management decisions (family reluctance to start corticosteroids, need for dose modifications) and must be discussed with families. There was general agreement about the side effect profiles of daily corticosteroids (S2 Table). When asked to compare a daily corticosteroid (DP or DD) with a weekend regimen (WP), panel members either offered no opinion or believed that weekend corticosteroids had fewer side effects, with specific comparisons shown in S2 Table.

## Eteplirsen

Statements and responses regarding use of eteplirsen are summarized in S3 Table and key statements are summarized in Fig 2. All experts agreed or strongly agreed that eteplirsen should be offered to those with an appropriate mutation provided there is a potential for clinical benefit based on disease stage (S3 Table, Statement 1a). Thirteen agreed or strongly agreed that eteplirsen is likely to slow disease progression (S3 Table, Statement 2b) and a majority (twelve) agreed or strongly agreed that eteplirsen slows respiratory decline (S3 Table, Statement 6a). Eight had no opinion regarding the effect on cardiac function (the remainder divided on likely benefit) presented in S3 Table, Statement 6b.

We then explored the panel members' opinions about potential benefit from eteplirsen relative to stage of disease. All would offer eteplirsen at diagnosis (S3 Table, Statement 4a), and most (12/15) would offer eteplirsen to patients with significant disability but maintaining some independent functioning (wheelchair dependent but still able to feed himself) (S3 Table, Statement 4d). The majority (fourteen), disagree with offering eteplirsen to a patient who is completely dependent (on ventilation and cannot drive a power wheelchair independently) (S3 Table, Statement 4e).

All panel experts agreed that eteplirsen could be given with corticosteroids (S3 Table, Statement 3a&b) and that it is appropriate to offer eteplirsen to patients not yet ready to start corticosteroids (S3 Table, Statement 8).

There was consensus (13 agreed or strongly agreed) that those on eteplirsen could be monitored by clinical measures such as motor outcomes, motor milestones, and pulmonary function tests (S3 Table, Statement 5b). No panelist agreed with a statement that muscle biopsy should be done to monitor effect of eteplirsen (S3 Table, Statement 5a).

There was universal agreement by all panel members that potential risks of weekly infusions of eteplirsen should be explicitly discussed with families prior to starting the drug and also full

agreement that it is well tolerated outside of risks associated with frequent infusions (S3 Table Statement 7).

## Discussion

The management of DMD is complex and requires a multidisciplinary approach. One aspect of care is medical therapy directed at preserving or improving muscle function. The recent care considerations are in agreement that corticosteroids are indicated for the medical management of DMD, but when those guidelines were developed, deflazacort was not yet FDA approved [7]. In addition, while the care considerations cited the available literature about intermittent corticosteroid dosing, there were no recommendations regarding the use of this regimen in practice. Similarly, eteplirsen was approved by the FDA after the CDC sponsored care considerations were developed [8–10]. The objective of this study was to collect the opinions of clinicians actively caring for patients with DMD regarding these treatments of muscle weakness and to determine where there is consensus on use of these treatments to aid clinicians in medical decision making. The information presented here is also expected to inform payers about current standards of care.

With two corticosteroid drugs used in DMD and daily or intermittent dosing schedules, clinical decision-making regarding corticosteroids has become more complex. Our data indicate that elements driving provider choice of corticosteroid regimen include effectiveness, side effect profile, cost to family, and payer requirements, all of which also likely contribute to individual family preferences. There continues to be limited data directly comparing effectiveness of different steroid regimens. Therefore, our methods did not allow for a clear expert opinion on relative effectiveness between all regimens. For example, those who disagreed with the statement "DP is more effective than DD" might have disagreed because they believe the two drugs have equal efficacy or that they feel DP is less effective. Daily corticosteroids are associated with multiple adverse effects (AEs) that have been well documented [7]. The consensus estimate of relative severity of side effects with different steroid regimens is summarized in Table 2, which might be of use in discussing the relative merits and side effects of each regimen with families. Our data suggest that payer requirements (rather than medical preference) influences the choice of corticosteroid for some clinicians and if payer requirements are not considered, DD or WP are preferred over DP.

Even before approval of new medicines for DMD, the reported cost associated with DMD management was high with an estimated mean annual out-of-pocket cost of $14,390 ($10,330-$22,970) in the US [11]. With FDA approval of new, very expensive treatments, cost for medication is an important part of DMD care. Prednisone is a generic medication and is far less expensive than deflazacort, although the cost to the family will be dependent on insurance

**Table 2. Consensus comparison of corticosteroid side effect profiles.**

| Factors | Daily prednisone | Daily deflazacort | Weekend Prednisone |
|---|---|---|---|
| Obesity / Weight gain | ◆◆◆ | ◆◆ | ◆ |
| Osteoporosis | ◆◆◆ | ◆◆◆ | ◆ |
| Behavior disturbance | ◆◆◆ | ◆◆ | ◆ |
| Delayed puberty | ◆◆◆ | ◆◆◆ | ◆ |
| Cataract | ◆◆ | ◆◆◆ | ◆ |
| Hypertension | ◆◆◆ | ◆◆◆ | ◆ |
| Adrenal Crisis | ◆◆◆ | ◆◆◆ | ◆ |

◆◆◆Highest risk; ◆◆Moderate risk; ◆lowest risk.

coverage. All but one of the expert panel felt that cost to the family should be explored when considering treatment options.

Eteplirsen is an antisense oligonucleotide designed to promote skipping of exon 51 in the *DMD* gene, thus restoring reading frame in patients with appropriate mutations [12]. Eteplirsen was granted accelerated approval by the FDA in September 2016, based on increased dystrophin level in muscle [13]. The clinical utility of the eteplirsen is less well documented and the results of the phase 3 trial have not yet been published, however clinicians need to make decisions about its appropriate clinical use. Eteplirsen is reported to be well tolerated over a period of 168 weeks duration [14]. However, discussion of eteplirsen should include the impact and inconvenience of weekly infusion, including the possible need for an implantable venous access port. We found consensus that eteplirsen should be offered to patients with confirmed mutations amenable to exon 51 skipping at diagnosis and through a stage of useful arm function, while it is not appropriate in very late disease.

Payers typically require regular, formal evaluation of the efficacy of eteplirsen for ongoing approval. Eteplirsen is expected to slow decline in muscle strength, not reverse disease course. There was a consensus that efficacy should be measured clinically by using standardized tests of motor and respiratory function, and long-term functional milestones such as loss of ambulation and need for respiratory support. The choice of specific motor function tests is dependent on the resources and physical environment in each clinic. For example, while the 6-minute walk distance (6MWD) has been used extensively in a research setting, it is often not practical in a busy clinic with crowded hallways [15,16]. There was complete agreement that muscle biopsy is not an appropriate measure of eteplirsen utility outside of research studies.

Since the completion of this work, the FDA gave accelerated approval to golodirsen, a medication designed to skip exon 53 in the *DMD* gene, through a mechanism similar to eteplirsen. Golodirsen is also administered by weekly infusions and has similar safety profile [7,17–20]. While it was not included in this study, clinical use (indications, risks and benefits, and monitoring) is likely to be similar to eteplirsen.

The members of the expert panel all manage patients with DMD in clinic and are having discussion of medical management with patients routinely. The work presented here reflects opinions regarding real world experiences and approaches to patient management. While there are some areas of strong consensus, there are other areas where consensus is lacking or most respondents neither agree nor disagree with the statement. The relative effectiveness of different corticosteroid regimens is one area where there is a clear need for more data.

The members of the panel participate in previous and ongoing clinical trials, including trials sponsored by companies whose drugs are discussed here. We note that expertise in clinical management is a pre-requisite for both expert opinion and clinical trial participation, so this overlap is not surprising. The authors have no personal benefit from use of the drugs discussed and affirm that the opinions provided are free of commercial bias to the extent possible.

Treatment options for DMD are rapidly evolving, with three new drugs approved in the past three years and evidence growing about use of older drugs. Comprehensive care guidelines are a critical basis for care, but the pace of drug development requires regular updates. The data presented here are intended to assist clinicians, families, and payers in decision making about medical treatment of muscle weakness in DMD while we await additional data. We anticipate that efforts to understand consensus with broad input similar to this effort using the Delphi method will be helpful as new treatments are approved, clinical trial data are released, and real world experience matures.

## Supporting information

**S1 Fig. PRISMA flow diagram of systematic literature searches.**
(TIF)

**S1 Table. Consensus statements & level of agreement (survey round 1).**
(DOCX)

**S2 Table. Consensus statements & level of agreement for treatment with corticosteroids (survey round 2&4).**
(DOCX)

**S3 Table. Consensus statements & level of agreement for treatment with eteplirsen (survey round 2&4).**
(DOCX)

**S1 File. DMD—Round 1.**
(XLSX)

**S2 File. DMD—Round 2.**
(XLSX)

**S3 File. DMD—Round 3.**
(XLSX)

## Acknowledgments

We would like to acknowledge Robert J Rogers for his assistance during the conduct of survey using an online portal "Survey Monkey".

## Author Contributions

**Data curation:** Sumit K. Jhamb.

**Formal analysis:** Sumit K. Jhamb.

**Investigation:** Hoda Z. Abdel-Hamid, Gyula Acsadi, John Brandsema, Emma Ciafaloni, Basil T. Darras, Susan T. Iannaccone, Chamindra G. Konersman, Nancy L. Kuntz, Craig M. McDonald, Julie A. Parsons, Carolina Tesi Rocha, Craig M. Zaidman, Russell J. Butterfield, Anne M. Connolly, Katherine D. Mathews.

**Methodology:** Hoda Z. Abdel-Hamid, Gyula Acsadi, John Brandsema, Emma Ciafaloni, Basil T. Darras, Susan T. Iannaccone, Chamindra G. Konersman, Nancy L. Kuntz, Craig M. McDonald, Julie A. Parsons, Carolina Tesi Rocha, Craig M. Zaidman, Russell J. Butterfield, Anne M. Connolly, Katherine D. Mathews.

**Project administration:** Sarah R. Rivera.

**Supervision:** Sarah R. Rivera, Katherine D. Mathews.

**Validation:** Russell J. Butterfield, Anne M. Connolly, Katherine D. Mathews.

**Writing – original draft:** Sarah R. Rivera, Sumit K. Jhamb.

**Writing – review & editing:** Sarah R. Rivera, Sumit K. Jhamb, Russell J. Butterfield, Anne M. Connolly, Katherine D. Mathews.

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
