## [Decision Letter · Decision Letter 0]

24 Aug 2020

PONE-D-20-22472

Medical management of muscle weakness in Duchenne muscular dystrophy

PLOS ONE

Dear Dr. Rivera,

Thank you for submitting your manuscript to PLOS ONE. After careful consideration, we feel that it has merit but does not fully meet PLOS ONE’s publication criteria as it currently stands. Therefore, we invite you to submit a revised version of the manuscript that addresses the points raised during the review process.

We look forward to receiving your revised manuscript.

Kind regards,

Alfio Spina, M.D.

Academic Editor

PLOS ONE

Journal Requirements:

'The organization in which SRR and SKJ are employed received payment from Sarepta to support the development of this manuscript. SRR and SKJ were in the role of facilitator, literature review coordinator, and medical writer to develop a manuscript based on survey outcomes, recommendations, and conclusion as reported by steering committee. They did not have any active role in experts panel selection, methodology development, interview questions finalization, outcomes assessment, and developing conclusion. Therefore, they had no influence on the decisions or outcomes of the entire program.

None of the others authors that served as expert panel and steering committee members were compensated in any way.

Also note that Sarepta had no influence on the decisions or outcomes of the entire program or any part of the manuscript.'

We note that you received funding from a commercial source: Sarepta

We also note that one or more of the authors are employed by a commercial company: Optum

C. Please also declare this commercial affiliation in your updated Competing Interests Statement, along with any other relevant declarations relating to employment, consultancy, patents, products in development, or marketed products, etc.  

Reviewers' comments:

Reviewer's Responses to Questions

**Comments to the Author**

1. Is the manuscript technically sound, and do the data support the conclusions?

Reviewer #1: Yes

Reviewer #2: Yes

2. Has the statistical analysis been performed appropriately and rigorously? 

Reviewer #1: Yes

Reviewer #2: Yes

3. Have the authors made all data underlying the findings in their manuscript fully available?

Reviewer #1: Yes

Reviewer #2: Yes

4. Is the manuscript presented in an intelligible fashion and written in standard English?

Reviewer #1: Yes

Reviewer #2: Yes

5. Review Comments to the Author

Reviewer #1: I thank the authors for their laborious work. I would suggest adding more questions in the Delphi iteration questionnaire regarding the motor power and pulmonary function tests in addition to the cardiac function in the field of steroid response as a comparison among DP, WP and DD, in order to yield the questionnaire more objective and semi quantitative. Triangulated research is needed in the future recommendation.

Reviewer #2: The idea of this study is important and the manuscript is well written.

The authors focus on a major gap in current treatment guidence for the medical management of musce weakness in DMD.

The topic, the data and the approach are interesting to neuroogy community. It shoud bu accepted without revision

6. PLOS authors have the option to publish the peer review history of their article (what does this mean?). If published, this will include your full peer review and any attached files.

Reviewer #1: **Yes: **MM

Reviewer #2: **Yes: **Bilgehan Atılgan ACAR

---

## [Author Response · Author response to Decision Letter 0]

27 Aug 2020

Thank you for the review and feedback for our manuscript, “Medical management of muscle weakness in Duchenne muscular dystrophy.” We have prepared all the requested additional documents and made all changes requested. 

This includes the following:

1. Both a marked-up and unmarked copy of the manuscript. You will find we made only minor changes to align with the PLOS ONE style requirements. 

2. We have uploaded all the files that contain our raw survey data as Supporting Information. There are no ethical or legal restrictions for the sharing of this data we were just unsure how it was to be shared in the initial submission. We apologize for this error. 

3. Within our revised cover letter, we have asked that our Funding Disclosure be updated to reflect the statements that better define the employment relationship with Optum for authors SRR and SKJ. 

4. Along these same lines we have also requested in the cover letter, an update to the Conflicting Interests Statement to better clarify the affiliation and support provided by both Sarepta and Optum.

---

## [Editor Report · Decision Letter 1]

1 Oct 2020

Medical management of muscle weakness in Duchenne muscular dystrophy

PONE-D-20-22472R1

Dear Dr. Sarah Rivera,

We’re pleased to inform you that your manuscript has been judged scientifically suitable for publication and will be formally accepted for publication once it meets all outstanding technical requirements.

Kind regards,

Alfio Spina, M.D.

Academic Editor

PLOS ONE
---

## [Editor Report · Acceptance letter]

6 Oct 2020

PONE-D-20-22472R1 

Medical management of muscle weakness in Duchenne muscular dystrophy 

Dear Dr. Rivera:

I'm pleased to inform you that your manuscript has been deemed suitable for publication in PLOS ONE. Congratulations! Your manuscript is now with our production department. 

Kind regards, 

on behalf of

Dr. Alfio Spina 

Academic Editor

PLOS ONE